# Cigarette Smoke Condensate Exposure Induces Receptor for Advanced Glycation End-Products (RAGE)-Dependent Sterile Inflammation in Amniotic Epithelial Cells

**DOI:** 10.3390/ijms22158345

**Published:** 2021-08-03

**Authors:** Helena Choltus, Régine Minet-Quinard, Corinne Belville, Julie Durif, Denis Gallot, Loic Blanchon, Vincent Sapin

**Affiliations:** 1CNRS, INSERM, GReD, Université Clermont Auvergne, 63000 Clermont-Ferrand, France; helena.choltus@uca.fr (H.C.); rquinard@chu-clermontferrand.fr (R.M.-Q.); corinne.belville@uca.fr (C.B.); loic.blanchon@uca.fr (L.B.); 2CHU de Clermont-Ferrand, Biochemistry and Molecular Genetic Department, 63000 Clermont-Ferrand, France; j_durif@chu-clermontferrand.fr; 3CHU de Clermont-Ferrand, Obstetrics and Gynecology Department, 63000 Clermont-Ferrand, France; denis.gallot@uca.fr

**Keywords:** maternal smoking, fetal membranes, RAGE, pPROM

## Abstract

Maternal smoking is a risk factor of preterm prelabor rupture of the fetal membranes (pPROM), which is responsible for 30% of preterm births worldwide. Cigarettes induce oxidative stress and inflammation, mechanisms both implicated in fetal membranes (FM) weakening. We hypothesized that the receptor for advanced glycation end-products (RAGE) and its ligands can result in cigarette-dependent inflammation. FM explants and amniotic epithelial cells (AECs) were treated with cigarette smoke condensate (CSC), combined or not with RAGE antagonist peptide (RAP), an inhibitor of RAGE. Cell suffering was evaluated by measuring lactate dehydrogenase (LDH) medium-release. Extracellular HMGB1 (a RAGE ligand) release by amnion and choriodecidua explants were checked by western blot. NF-κB pathway induction was determined by a luciferase gene reporter assay, and inflammation was evaluated by cytokine RT-qPCR and protein quantification. Gelatinase activity was assessed using a specific assay. CSC induced cell suffering and HMGB1 secretion only in the amnion, which is directly associated with a RAGE-dependent response. CSC also affected AECs by inducing inflammation (cytokine release and NFκB activation) and gelatinase activity through RAGE engagement, which was linked to an increase in extracellular matrix degradation. This RAGE dependent CSC-induced inflammation associated with an increase of gelatinase activity could explain a pathological FM weakening directly linked to pPROM.

## 1. Introduction

Worldwide, between 15 and 20% of pregnant women still smoke, even though smoking cigarettes is a well-known risk factor for preterm prelabor rupture of fetal membranes (pPROM). This obstetrical complication can lead to disaster for both the mother and her baby, such as chorioamnionitis or oligohydramnios, and it is responsible for 30% of preterm births [1,2,3,4,5]. How does maternal smoking induce pPROM? Current data support the hypothesis that pPROM is caused most often by the exacerbation of the same cellular and molecular mechanisms that lead to term rupture: apoptosis, oxidative stress, extracellular matrix degradation, senescence, and sterile inflammation. In vitro studies offer different possibilities for tabagism modelling. First, some studies use a specific component of cigarettes; for example, nicotine. A recent study highlighted the dose-dependent impact of nicotine on maternal and fetal inflammatory responses in pregnant rats [6]. Moreover, two products allow the testing of cigarette smoke’s impact regarding its whole composition [7]. The first, which samples the water-soluble compounds, is cigarette smoke extract (CSE), which is used primarily as an inducer of oxidative stress (OS) [8,9,10,11,12,13]. The second, cigarette smoke condensate (CSC), is obtained from a particular part of cigarette smoke. CSE has already shown that cigarette smoke induces OS, senescence, and apoptosis, and also activates a p38 MAPK inflammatory cell pathway in FM cells, mechanisms that are implicated in tissue weakening [9,13]. Bredeson et al. also demonstrated that CSE induces high-mobility group B1 (HMGB1) release, a protein that acts as an alarmin when it becomes extracellular and activates sterile inflammation [10]. Moreover, in vivo, CSE injections also induce senescence and inflammation (the release of interleukin in amniotic fluid) in mice [8]. CSC demonstrated that it can dysregulate gene expression (for example, those targeted by a retinoid pathway) in amniotic cells [14], but the available studies on CSC implications remain scarce.

To date, the signaling actors and pathways activated in these tissues by cigarette smoke exposure are poorly understood. Nevertheless, mounting evidence supports the role of the receptor for advanced glycation end-products (RAGE) in inflammatory diseases associated with tabagism. Tobacco smoke has been shown to initiate the non-enzymatic Maillard reaction, leading to the formation of advanced glycation end-products (AGEs), which are one of the main ligands of RAGE. Moreover, cigarette smoke exposure has also been associated in vivo with the release or overexpression of HMGB1 or S100 proteins (A6, A9) in lungs [15,16,17,18,19]. Most of all, the abortion of RAGE activity in mice and humans has been described as being associated with a decrease of cigarette smoke-induced inflammation in different tissues. This is the case at the pulmonary level [20,21,22], in cardiovascular diseases [23], and for intrauterine growth restriction (IUGR) syndrome in the obstetrical sphere [24]. RAGE is a 50 kDa cell membrane receptor that is implicated in many cell responses, including inflammation from its interaction with alarmins (HMGB1 and AGEs). Following its engagement through ligand fixation, RAGE can activate many pro-inflammatory cell-signaling pathways, such as NFκB or MAPK [25,26]. All previous data led to our hypothesis that cigarette smoking, modeled by CSC use and focusing on RAGE activation, induces FM weakening by increasing sterile inflammation.

## 2. Results

### 2.1. Cigarette-Smoke Condensate Induces a Danger Response in the Fetal Amnion and Its Cells

The impact of CSC treatment on cell toxicity was evaluated in FM explants (dissociated amnion and choriodecidua) by measuring LDH release in culture media (Figure 1A). Here, CSC provoked significant cell toxicity (ratio of 2.9) only in the amnion layer. Further, the release of the alarmin HMGB1 was investigated, and here again, we noted the induction of such a release only in the amnion layer (ratio of 6.6) but not in the choriodecidua layer (ratio of 0.9) (Figure 1B). Considering these results, further investigation of what happens in the amnion after CSC exposure was conducted by focusing on amniotic epithelial cells (AECs), which are in direct contact with the amniotic fluid that contains cigarette metabolites. Therefore, for the entire amnion layer, we demonstrated that CSC induced amniotic cell LDH release (ratio of 2.0) (Figure 1C) and provoked the nuclear toward cytosolic HMGB1 translocation. Indeed, whereas the control (DMSO, dimethylsulfoxyde) exhibited classical HMGB1 nuclear localization, CSC-treated cells showed a cytosolic HMGB1 cloud, a key step in subsequent extracellular release (Figure 1D).

### 2.2. RAGE Axis Actors Are Expressed in Primary Amniotic Epithelial Cells (pAECS)

HMGB1 is a well-known ligand of RAGE, implicated both in pro-inflammatory processes. It was necessary to confirm the expression of RAGE axis actors, including RAGE itself, as well as three other proteins: TIR adaptor protein (TIRAP), Myeloid differentiation primary response 88 (MyD88), and Diaphanous-1. In fact, many studies have established that RAGE requires one of these intracellular adaptors to induce a cell signal [27,28,29]. As shown in Figure 2, the RT-PCR demonstrated the expression of RAGE and its three adaptors in pAECs (2A). Then, immunocytochemistry experiments (2B) confirmed the protein expression of these RAGE axis actors. Further, we can observe that RAGE is localized in the nuclei and at the cell membrane of pAECs, in accordance with its functions as a membrane receptor and as a regulator of DNA repair [25,30].

### 2.3. Cigarette Smoke Condensate Induces RAGE Signaling Cascade in pAECs

The pAECs were treated for 48 h with either only CSC or with a RAGE competitive inhibitor, RAP. RT-qPCR was performed on pAECs to quantify RAGE axis members’ expression. The results revealed no impact on RAGE, Dia-1, or TIRAP expression. However, CSC significantly increased the Myd88 transcription level (ratio of 1.6), and this augmentation was lower in the case of co-treatment with RAP (ratio of 1.3) (Figure 3A). Yet, the NFκB luciferase reporter assay demonstrated that pAECs exposed to CSC activated the pro-inflammatory NFκB cell signaling (ratio of 2.4), which was aborted in the presence of RAP (ratio of 1.7) (Figure 3B). Furthermore, the immunofluorescence of p65 NFκB protein supported the idea that CSC activates this pathway by passing through the RAGE receptor. As described in the literature, control cells exhibited p65 localization mainly in cytosol (a cell compartment where p65 is mainly sequestered when there is no activation). Linked to the signal transduction by RAGE (blocked by RAP), the CSC treatment affected this localization with an addressing of p65 from the cytosol to the perinuclei/nuclei, which attests NFκB activation (Figure 3C).

### 2.4. Cigarette Smoke Condensate Induces Pro-Inflammatory Response in pAECs

As CSC engages the RAGE pathway, one of the most well-known consequences of RAGE activation to study is the increasing production of pro-inflammatory mediators, such as cytokines. In fact, transcription quantification of pro-inflammatory cytokines demonstrated a significant fold change induction of IL1β (ratio of 10.13), IL8 (ratio of 904.3), and IL6 (ratio of 291.10^3^), but not TNFα (data not shown), after CSC treatment. Above all, RAGE inhibition by RAP led to an abortion of IL8 induction (ratio of 45.1) and a smaller increase of IL6 expression (ratio of 142.10^3^), but it had no impact on the IL1β fold change (ratio of 13.33) (Figure 4A). The cytokines released by pAECs were investigated after 48 or 72 h of CSC treatment, using ELLA assays (Figure 4B). First, we observed that CSC provoked a release of IL1β (ratio of 2.0 at 48 h), IL8 (ratio of 4.3 at 48 h and 13.6 at 72 h), and IL6 (ratio of 2.7 at 72 h). Second, these results confirmed the inhibitory effect of RAP on CSC inflammatory action (as obtained before) for IL8 (ratio of 6.1) and IL6 (ratio of 1.6) release at 72 h, but also for IL1β at 48 h (ratio of 1.8).

FM integrity relies on an extracellular matrix (ECM) structure. Some proteins, such as the metalloproteases 2 and 9, exercise gelatinase activity, leading to ECM degradation. We investigated whether CSC treatment could have an impact on this by first modulating the transcription of MMP2 and MMP9, and then increasing gelatinase activity. The RT-qPCR experiment revealed a significant induction of MMP9 expression (ratio of 8.5) by CSC at 48 h, but no impact on the MMP2 expression level (ratio of 0.7) (Figure 5A). We also observed that RAP decreased MMP9 induction (ratio of 4.8) and diminished MMP2 expression compared to the control (ratio of 0.6). Then, the measurement of gelatinase activity after 48 or 72 h of CSC treatment, whether or not it was combined with RAP, revealed the induction of gelatinase activity at 72 h, compared to the control (ratio of 1.14); this induction was suppressed in the presence of the RAGE inhibitor RAP (ratio of 0.94) (Figure 5B).

## 3. Discussion

Many pathologies are caused by cigarette-induced sterile inflammation, which can lead to lung impairments (e.g., chronic obstructive pulmonary disease), cancer mechanisms (in lung, larynx, and pancreas), cardiometabolic abnormalities (e.g., atherosclerosis, dyslipidemia, hyperinsulinemia, and cardiomyopathy), as well as joint (osteoarthritis) or maternal-fetal disorders [2]. More and more studies highlight the implication of RAGE signaling in the establishment of sterile inflammatory diseases that are linked with exposure to primary and secondhand cigarette smoke. For many decades, it has been well known that cigarette smoking has deleterious effects on pregnancy outcomes, such as preterm births and pPROM [31,32,33]. The involvement of RAGE has already been described in IUGR development, and it is also well known that inflammation is implicated in physiopathological FM weakening. Surprisingly, only a few studies have investigated the cellular signaling actors responsible for cigarette-induced inflammation in pPROM pathogenesis. Nevertheless, it is important to underline that this inflammation is of a “sterile type”, involving the release of alarmins, which could be recognized by the RAGE receptor. From this initial evidence, our work consisted of investigating whether RAGE was implied in FM weakening and whether this is a new link between maternal smoking and premature sterile inflammation.

Human FM are composed of two layers: the amnion and the choriodecidua. Our in vitro experiments identified a different response to CSC from each of these tissues. Indeed, we noticed a significant induction of cell toxicity and an induction of HMGB1 release only in the amnion. The choriodecidua did not seem to be sensitive to CSC treatment. This could be explained by the fact that in vivo cigarette compounds are classically accumulated in amniotic fluid and in direct contact with the amnion (particularly its epithelial cells), but this is not the case for the choriodecidua. Our work aimed to more precisely describe the inflammatory response activated by CSC at the cellular amniotic level. We demonstrated that cigarette smoke compounds induce cell toxicity and provoke a release of alarmins (HMGB1), which is associated with sterile inflammation in pAECs. The release of DAMPs has already been identified in other models (e.g., chronic obstructive pulmonary disease), where it is well established that cigarette smoke exposure can induce necrotic death in airway epithelial cells, provoking DAMP (HMGB1, AGEs, S100, HSP70, and dsDNA) accumulation in the extracellular space [18,34,35,36]. These DAMPs are recognized by pattern recognition receptors (PRRs), such as RAGE. The engagement of these receptors can lead to an inflammatory response through the activation of pro-inflammatory pathways, such as NFκB. RAGE expression has already been identified in human FM. Our team already reported that RAGE is overexpressed, not only in the amnion compared to choriodecidua, but also in the rupture zone compared to the intact one. In parallel, a previous study demonstrated that FM are sensitive to RAGE ligands: the AGEs and HMGB1 [37]. This preferential expression of RAGE in the amnion could be exacerbated by contact with smoking compounds, explaining such amniotic susceptibility to CSC.

RAGE is a transmembrane receptor implicated in many cell responses (e.g., apoptosis, proliferation, inflammation, and cell migration), but cannot act alone [25]. Indeed, RAGE is deficient in intrinsic activity and requires intracellular adaptors to induce cell-signaling cascades. In the literature, we found three major adaptor proteins that interacted with the RAGE cytosolic domain and were described in the inflammatory signal: TIRAP, MyD88 shared with the toll-like receptors TLR2 and TLR4, and Diaphanous-1 [28,29]. For the first time, RAGE and its three adaptors were identified in AECs. Interestingly, our work additionally revealed a significant overexpression of MyD88, a RAGE intracellular adaptor whose expression was decreased by RAGE inhibition. Some studies have already investigated the role of Myd88 regarding cellular consequences of smoking. For example, Myd88-dependent signaling regulates, at least partially, CS-induced inflammation in bronchial epithelial cells, supporting the role of RAGE [34]. Moreover, the Filipovich group demonstrated that knockout mice for MyD88 were protected against *Escherichia coli*-induced preterm delivery, proving the essential role of Myd88 in septic preterm delivery, which allowed us to suppose that it could also be implicated in sterile cases of preterm birth [38]. Then, our work demonstrated that RAGE is required for the activation of NF-κB, which is a pro-inflammatory pathway well-known for inducing the production of pro-inflammatory mediators, such as cytokines. NF-κB was already known to be implied in childbirth, since human labor is associated with a constant increase of NF-κB activity in the amnion [39]. Moreover, Sheller-Miller et al. demonstrated that the in vivo injection of an NF-κB inhibitor delays LPS-induced preterm delivery [40]. Thus, by its impact on this cell pathway, it seems that maternal smoking could provoke preterm birth by affecting other events of parturition (preterm labor) in addition to pPROM.

Finally, our work illustrated the downstream effects of this RAGE activation by CSC in pAECs following NF-κB activation. In this way, this study highlighted that CSC-treated pAECs exhibit higher levels of pro-inflammatory cytokine transcription and release (IL6, IL8, and IL1β). In addition, RAGE inhibition led to the decrease of IL1β and the cessation of IL8 and IL6 release. This directly proves that cigarette smoke (CS) can induce inflammation through a RAGE pathway. However, the absence of a total abortion for IL1β release could suggest the intervention of other actors, such as the well-known TLR4, which can also interact with alarmins. For example, it was found that nicotine modulates the expression of TLR2/4 into cord blood mononuclear cells [41]. Cheng et al. also demonstrated that HMGB1 translocation can mediate CS-induced pulmonary inflammation through the TLR4/MyD88 pathway [42]. However, Allam et al. demonstrated that RAGE and TLR4 differentially regulate airway responsiveness to cigarette smoke (CS). Indeed, the authors used RAGE and TLR4 knockout mice and found that only RAGE deletion procured protection against CS-induced neutrophils and airway responsiveness [43]. In our amniotic context, we tested at the beginning of the project the impact of CSC on TLR4 expression and found that there is no induction (ratio 1.1 by qRT-PCR, Appendix A) excluding the TLR4 hypothesis. Nevertheless, further studies are required to explore the potential cooperation between these two PRRs concerning sterile inflammation in FM.

Sterile inflammation is clearly associated with extracellular matrix (ECM) degradation, which is also a recurrent phenomenon that destructures tissue integrity, in fine, the FM weakening. The major actors in this ECM deterioration are enzymes called metalloproteinases. Two major groups comprise this family: gelatinases (MMP2 and 9) and collagenases (MMP1, 7, 8, and 13). Amniotic epithelium express MMP1, 2, and 9. Our work demonstrated that CSC stimulates MMP9 transcription in amniotic epithelial cells but has no impact on MMP2 and MMP1 (data not shown for MMP1). Moreover, the RAGE blockade led to a decrease in CSC-induced MMP9 expression and the physiological expression of MMP2. We also investigated gelatinase activity on cell supernatants by zymography assay. Here, we revealed an induction of gelatinase activity by CSC at 72 h, and this induction was aborted by RAP treatment. In the literature, RAGE was implicated in CS-induced MMP expression and/or activity (MMP2, 9, and 14) in the respiratory tree [44]. Moreover, MMP9 was revealed to be increased in amniotic fluid and fetal plasma in the case of pPROM; thus, it is considered a potential biomarker of pPROM [45,46]. Polymorphism in the MMP9 promoter is associated with an increased risk of pPROM [47]. Finally, many studies have described the implication of NFκB pathway in MMP9 upregulation in the case of inflammation [48,49,50,51]. Our results revealed a new way in which CSC weakens FM through RAGE and is in total accordance with the results obtained in other tissues.

In conclusion, our work makes clear a way that cigarette smoking (CSC) can induce FM weakening and, in fine, a pPROM, as summarized in our RAGE-dependent model of action (Figure 6).

Indeed, we proved that CSC provokes a sterile inflammatory response in amnion epithelial cells by activating the RAGE pathway. This increase in inflammation may accelerate the physiological weakening of FM. Due to the complexity of the molecules, as well as the gas released during cigarette smoking, it would be interesting to compare the effects of CSC (the particular phase) to the effects of CSE (sampling the water-soluble compounds) and to check the activation and implication of RAGE pathways in such a mixed exposure case. Nevertheless, this study still supports a cellular explanation for the negative effects of tobacco smoke on pregnant women and paves the way for developing new preventive strategies to protect smoking mothers from pPROM by targeting the RAGE pathway.

## 4. Materials and Methods

### 4.1. Chemicals

CSC (20 mg/mL in dimethylsulfoxyde (DMSO)) was purchased from Kentucky Tobacco & Research Development Center (Lexington, KY, USA), and DMSO was purchased from Sigma-Aldrich (Saint-Quentin-Fallavier, France). RAGE Antagonist Peptide (RAP) (Cat N°553031, Calbiochem^®^, 5 mg/mL in water) was obtained from Sigma-Aldrich (Saint-Quentin-Fallavier, France). The cell culture medium and antibiotics (streptomycin, penicillin, and amphotericin B) were obtained from Fisher Scientific™ (Illkirch-Graffenstaden, France), and fetal bovine serum was purchased from Eurobio Scientific (Les Ulis, France). Collagen I (3 mg/mL) was obtained from Stemcell Technologies (Grenoble, France). Superscript IV First-Strand-Synthesis System, Taq DNA polymerase recombinant (10342020), and Pierce BCA Protein Assay Kit (23225) were obtained from Fisher Scientific™ (Illkirch-Graffenstaden, France) and SYBR Green from Roche (Meylan, France).

### 4.2. Tissue Collection

Full-term FMs were collected from non-smoking women with healthy pregnancies from vaginal or scheduled cesarean deliveries (e.g., breech presentation or scarred wombs) (Centre Hospitalier Universitaire Estaing, Clermont-Ferrand, France) after obtaining informed consent. Gestational ages were 39.04 ± 0.38 weeks, the mean maternal ages were 31.26 ± 3.79 years, and the maternal body mass index was 26.33 ± 4.14. The selected FMs were collected from singleton pregnant women who had no underlying diseases, gestational diabetes, or clinical chorioamnionitis (defined by maternal fever, uterine tenderness, and/or purulent amniotic fluid). The research protocol was approved by the institutional regional ethics committee (DC-2008-558).

### 4.3. Tissue and Cell Culture

Explants (dissociated) of the amnion and choriodecidua were cultivated (5% CO_2_; 95% humidified air; 37 °C) in Dulbecco’s modified eagle medium/nutrient mixture F-12 (DMEMF12- GlutaMAX, Gibco™, Illkirch-Graffenstaden, France), supplemented with 10% FBS, 100 mg/mL of streptomycin, 100 U/mL of ampicillin, and 25 mg/mL amphotericin B (Gibco™, Illkirch-Graffenstaden, France). Explants were 2 cm^2^ in size, cut 2 cm away from the preplacental edge, and prepared by dissection. Tissue fragments were transferred (in duplicate) to 24-well culture plates and incubated in the cell media at 37 °C for 1 h before treatment.

pAECs were cultivated under standard conditions (5% CO_2_; 95% humidified air; 37 °C) in Dulbecco’s Modified Eagle Medium F-12 nutrient mixture (DMEM-F12- GlutaMAX) supplemented with 10% FBS, 100 µg/mL/mL of streptomycin, 100 U/mL of ampicillin, and 25 µg/mL amphotericin B. As previously validated, the isolation of pAECs was conducted in three trypsinization steps (10, 20, and 30 min), followed by the scraping of the amnion [52]. Cells were filtered to remove the collagen, centrifuged for 5 min at 1000 rpm, and grown on culture dishes coated with collagen I (1/50 dilution in PBS 1X) in complete media.

### 4.4. Tissue and Cell Treatment

Cells or explants were treated with DMSO or CSC (respectively 100 or 500 µg/mL) in the absence or presence of RAP (12.7 or 63.6 µg/mL) for 24 h, 48 h, or 72 h (with treatment renewal at 48 h).

### 4.5. Global Cellular Distress Determination

To evaluate the impact of treatment on global cell suffering, the release of the intracellular enzyme lactate dehydrogenase (LDH) into the cell media was quantified on an automate (Siemens Vista, Paris, France) using an enzymatic assay, following the manufacturer’s recommendations.

### 4.6. Western Blot Analysis of HMGB1 Release

First, the supernatants of the treated explants were concentrated into 2 kDa centrifugal filter units (Vivacon^®^500, Sartorius, Aubagne, France) for protein concentration and purification, following the manufacturer’s instructions. Then, the proteins were resolved on a 4–15% Mini-PROTEAN^®^ TGX Stain-Free™ Precast Gel (Bio-Rad, Marnes-la-Coquette, France) to perform total protein normalization [53]. Before the transfer, stain-free imaging was completed. This technology utilizes a proprietary trihalo compound to enhance natural protein fluorescence by covalently binding to tryptophan residues with brief UV activation (Bio-Rad, Marnes-la-Coquette, France). Then, the transfer was performed on a nitrocellulose membrane (Bio-Rad, Marnes-la-Coquette, France) and saturated for more than 90 min with 5% skimmed milk in a Tris-Buffered Saline (TBS) 1X. HMGB1 primary antibody (1/10000, ab79823, Abcam, Paris, France) was diluted in 5% skimmed milk-TBS 1X-TWEEN^®^20 0.1% and incubated overnight at 4 °C. The next day, the membrane was washed three times with TBS/TWEEN^®^20 0.1% and incubated at room temperature with a horseradish peroxidase coupled with a secondary antibody anti-rabbit (1/10000, BI 2407, Abliance, Compiègne, France) for 90 min. The revelation was completed using an ECL Clarity kit for a western blot on the ChemiDoc™ imaging system (Bio-Rad, Marnes-la-Coquette, France). Image Lab Software (Bio-Rad, Marnes-la-Coquette, France) was used for quantification. Results are expressed as a mean of at least three independent experiments.

### 4.7. RT-PCR and Quantitative RT-PCR on Explants and Cells

After the disruption step with a Precellys homogenizer (Bertin Technologies, Montigny-le-Bretonneux, France) using ceramic beads (KT03961, Ozyme, Saint-Cyr-l’École, France), the total number of RNAs were extracted from the explants using RNAzol^®^ RT (RN190, Molecular Research Center, Cincinnati, OH, USA). For the cells, RNA extraction was performed with NucleoSpin RNA (REF 740955.50, Macherey-Nagel, Hoerdt, France), following the manufacturer’s instructions. Reverse transcription was realized on 1 μg of RNA using a Superscript IV First-Strand-Synthesis System for reverse transcription polymerase chain reaction (RT-PCR). PCR experiments were performed using specific oligonucleotides (Table 1).

The results were analyzed on a 2% agarose gel. Gene expression was assessed by quantitative RT-PCR (RT-qPCR), which was performed using LightCycler^®^480 SYBR Green I Master (Roche, Meylan, France). Transcript quantification was performed in duplicate on at least four independent experiments. Standard curves were used to quantify the amount of amplified transcripts. Results were normalized to the geometric mean of the human housekeeping genes RPL0 (36b4) and RPS17 (acidic ribosomal phosphoprotein P0 and ribosomal protein S17, respectively), as recommended by the MIQE guidelines [54].

### 4.8. Cytokine Release Assay

The release of IL1β, IL6, and IL8 in the culture media was quantified after 48 h of CSC treatment using automated multiplex immunoassays on Ella™ (San Jose, CA, USA). Finally, cytokine concentrations were normalized to total protein concentration, and the fold change “treated/control” was reported.

### 4.9. Immunofluorescence

pAECs grown on collagen type I-coated coverslips in six-well plates were washed with PBS 1X before fixation with 4% paraformaldehyde (Electron Microscopy Sciences, 15710; can be stored at −80 °C). After permeabilization in PBS 1X/FBS 10%/Triton 0.1% for more than 90 min, the primary antibody against RAGE (1/1000, ab37647, Abcam, Paris, France), HMGB1 (1/400, ab79823, Abcam, Paris, France), Myd88 (1/250, ab133739, Abcam, Paris, France), TIRAP (1/100, ab17218, Abcam, Paris, France), Diaphanous-1 (1/400, ab11173, Abcam, Paris, France), and p65 NF-κB (1/400, 8242, Cell signaling, Saint-Cyr-L’Ecole, France) was applied overnight at 4 °C. After three washes in the permeabilization buffer, the secondary antibody, anti-rabbit Alexa Fluor^®^ 488 (1/1000, A21206, Life Technologies, Villebon-Sur-Yvette, France), was incubated for 2 h at room temperature. Slides were washed three times in PBS 1X and incubated with Hoescht (15 min, dilution in PBS 1X 1/10,000; bisBenzimide H, 33258, Sigma-Aldrich, Saint-Quentin-Fallavier, France). Finally, slides were mounted with CitiFluor™ Tris-MWL 4-88 (Electron Microscopy Science, Hatfield, PA, USA) and examined under a Zeiss LSM800 Airyscan for cells. For the negative controls, incubation without the primary antibody was performed.

### 4.10. NFκB Gene Reporter Luciferase Assay

pAECs grown at 80% confluence in six-well plates were co-transfected with 500 ng of pNF-κB-Firefly Luciferase plasmid (Agilent) expressing luciferase under control of NF-κB responsive elements and 50 ng of pRL-TK-Renilla-Luciferase expression vector (Promega, Charbonnières-les-Bains, France), which allowed the normalization of the transfection efficiency. Transfections were performed using 3.75 µL of Lipofectamine 3000 (Fisher Scientific, Illkirch-Graffenstaden, France) and 2 μL of reagent P3000 (Fisher Scientific, Illkirch-Graffenstaden, France) per well. Cells were treated with DMSO or CSC (100 µg/mL), whether combined or not with RAP (12.7 µg/mL), 24 h after the transfection for 48 h. Firefly and Renilla luciferase activities were then measured with a FB12 luminometer (Berthold, Thoiry, France) in the cellular extracts using the dual-luciferase reporter assay system (Promega, Charbonnières-les-Bains, France), according to the manufacturer’s instructions. This experiment was repeated three times (each condition in duplicate).

### 4.11. Gelatinase Activity Measurement

Gelatinase activity was measured on a cell culture media (1/500 diluted) using a gelatinase activity assay kit (MAK348, Sigma-Aldrich, Saint-Louis, MI, USA), following the manufacturer’s instructions. Fluorescence was measured in kinetic mode at 37 °C for two hours on a FlexStation microplate reader (Molecular devices, San Jose, CA, USA).

### 4.12. Statistical Analysis

The data expressed as the mean ± standard error of the mean are an average of duplicates or triplicates of at least three independent experiments. Since the results did not follow normal distribution (consequence of the small N), the comparison of means was performed by non-parametric tests (Mann–Whitney t-test or one-way ANOVA Kruskal–Wallis test, followed by multiple comparison with Dunn’s correction for the comparison of more than two groups) using PRISM software 5.02 (GraphPad Software Inc., San Diego, CA, USA). For all studies, values were considered significantly different at *p* < 0.05 (*), *p* < 0.01 (**), and *p* < 0.001 (***). Mean ratio (treated/control groups) values are indicated in the results part.

## Figures and Tables

**Figure 1 ijms-22-08345-f001:**
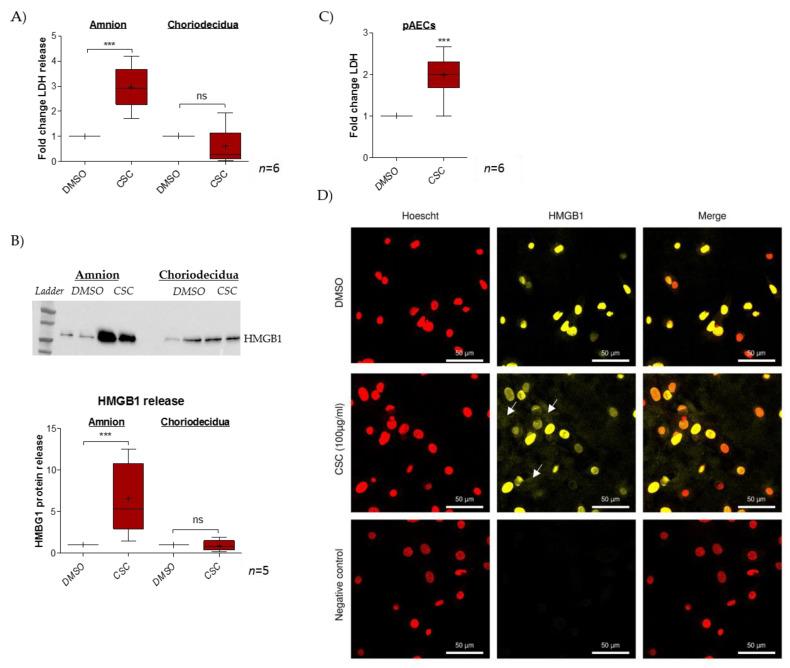
CSC treatment induces a cell danger response into amnion. (**A**) Cell toxicity was evaluated by measuring LDH release in a supernatant culture after a 24 h treatment with CSC in amnion and choriodecidua (*n* = 6). (**B**) HMGB1 release in a supernatant culture after a 48 h treatment with CSC was quantified by the western blot method in amnion and chori-odecidua; the results are reported in the histogram (*n* = 5). (**C**) Toxicity was evaluated by LDH release measurement in primary amniocytes (pAECs) cell-culture supernatants after a 48 h treatment with CSC (100 µg/mL) (*n* = 6). (**D**) HMGB1 nuclear toward cytosolic translocation was investigated by immunocytochemistry on pAECs after 7 h treatment with CSC (yellow staining; Alexa488). Nuclei were counterstained with Hoechst (red). Scales bars: 50 µM (magnification ×200). Negative control was realized without primary antibody hybridization. White arrows indicate the cytosolic cloud of HMGB1. Comparison with the control (DMSO) was realized by a Mann-Whitney *t*-test. *** means *p* < 0.001, and “ns” means “not significant”. Results are presented in Tukey boxes; means are indicated by “+”.

**Figure 2 ijms-22-08345-f002:**
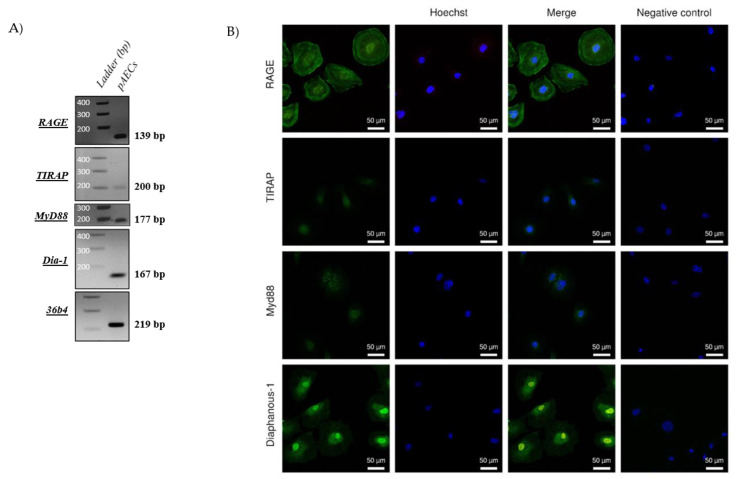
Amniotic epithelial cells expressed a functional RAGE axis. (**A**) RNA expressions of RAGE, TIRAP, Myd88, and Diaphanous-1 were detected by RT-PCR in pAECS. Negative controls were performed in the absence of a cDNA template. (**B**) The RAGE receptor and its adaptors, Diaphanous-1, MyD88, and TIRAP (green staining, Alexa488), were detected by immunocytochemistry on primary amniocytes (pAECs). Nuclei were counterstained with Hoechst (blue). Scales bars: 50 µM (magnification ×200). Negative control was realized without primary antibody hybridization.

**Figure 3 ijms-22-08345-f003:**
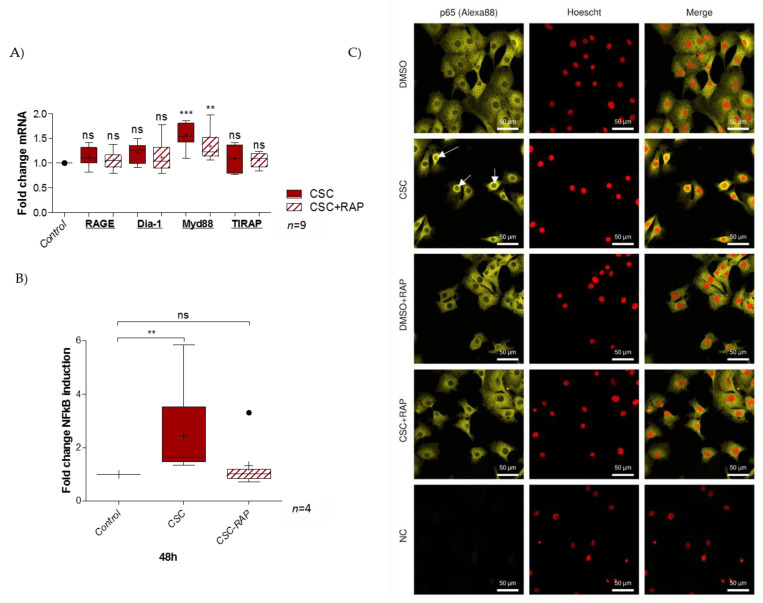
Activation of RAGE axis by CSC-response in pAECs. (**A**) Quantification of RAGE and its signaling adaptors (Diapahnous-1, Myd88, and TIRAP) transcription by RT-qPCR following 48 h of CSC (100 µg/mL) +/− RAP (12.7 µg/mL) treatment of pAECs (*n* = 7). (**B**) NFκB luciferase reporter assay was performed after 48 h of CSC treatment (100 µg/mL), whether combined or not with RAP (12.7 µg/mL) (*n* = 4). (**C**) p65-NFκB relocalization was investigated by immunocytochemistry on pAECs, whether treated or not with CSC for 48 h (yellow staining; Alexa488). Nuclei were counterstained with Hoechst (red). Scales bars: 50 µM (magnification ×200). Negative controls were realized without primary antibody hybridization. White arrows indicate perinuclear/nuclear relocalization of the p65 protein. A comparison of conditions was realized by a Kruskal–Wallis one-way ANOVA test, followed by a Dunn’s post-test. ** means *p* < 0.01, *** means *p* < 0.001, and “ns” means “not significant”. Results are presented in Tukey boxes, and means are indicated by “+”.

**Figure 4 ijms-22-08345-f004:**
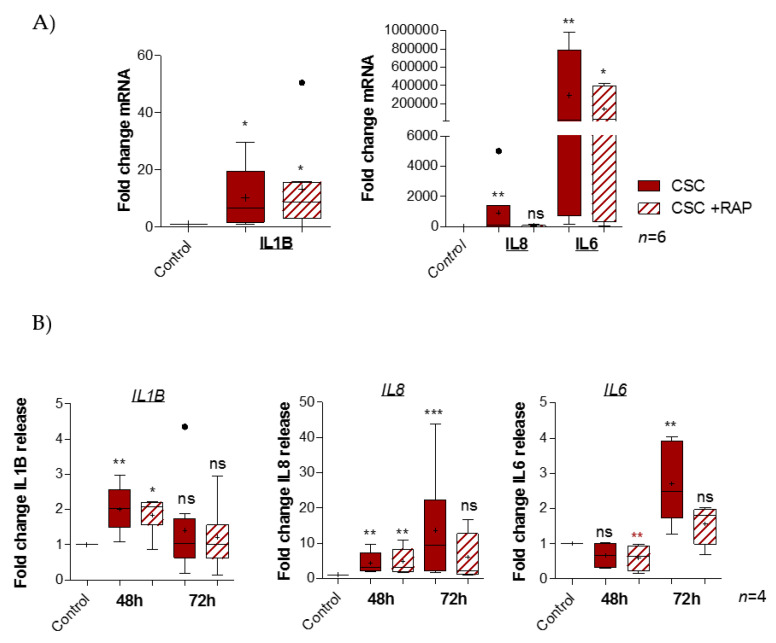
RAGE is implicated in CSC-induced pro-inflammatory cytokine production in pAECs. (**A**) Quantification of cytokine mRNA expression by RT-qPCR following 48 h of CSC (100 µg/mL) +/− RAP (12.7 µg/mL) treatment of pAECs (*n* = 6). (**B**) Pro-inflammatory cytokine (IL8, IL6, IL1β) secretion was quantified by ELLA technology after 48 and 72 h (with 48 h re-treatment) of CSC treatment (*n* = 4). A comparison of conditions was realized by a Kruskal–Wallis one-way ANOVA test, followed by a Dunn’s post-test. * means *p* < 0.05, ** means *p* < 0.01, *** means *p* < 0.001, and “ns” means “not significant”. Cigarette Smoke Condensate Enhances Gelatinase Activity in pAECs through the RAGE Pathway.

**Figure 5 ijms-22-08345-f005:**
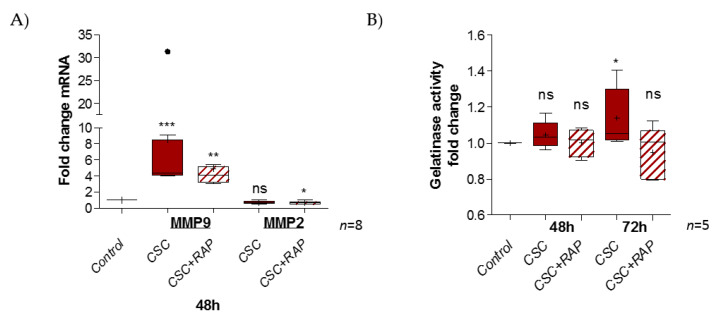
CSC exposure stimulates gelatinase activity in pAECs through the RAGE pathway. (**A**) Transcription of MMP2 and MMP9 gelatinases were measured by RT-qPCR into pAECS after 48 h of CSC treatment (100 µg/mL), whether combined or not with RAP (12.7 µg/mL) (*n* = 8). (**B**) Gelatinase activity was studied by a zymography kit assay on pAECs cell media after 48 and 72 h (with 48 h re-treatment) of CSC treatment (*n* = 5). Statistical analysis was performed using a Kruskal–Wallis one-way ANOVA test, followed by a Dunn’s post-test. * means *p* < 0.05, ** means *p* < 0.01, *** means *p* < 0.001, and “ns” means “not significant”. Results are presented in Tukey boxes, and means are indicated by “+”. Black dots indicate values outside of the Tukey boxes.

**Figure 6 ijms-22-08345-f006:**
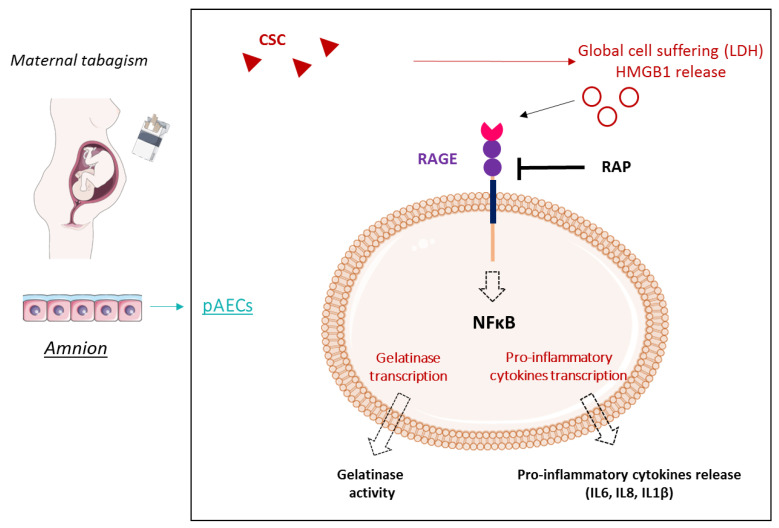
Maternal tabagism model of negative consequences on amniotic cells by the induction of RAGE-dependent sterile inflammation and gelatinase activity. Sterile inflammation is a key phenomenon of FM weakening, not only in physiological rupture, but also in pPROM. Exposure to tobacco during pregnancy is a well-known risk factor of pPROM. We demonstrated here that CSC induces an in vitro HMGB1 release by the amnion, a well-known danger signal. Then, this alarmin, a major ligand of RAGE, can induce a pro-inflammatory response (NF-κB activation and cytokine production) through the RAGE pathway in amniotic epithelial cells, which suggests the essential role of RAGE in FM rupture and pPROM. smart.servier.com was used to create the figure.

**Table 1 ijms-22-08345-t001:** Forward and reverse primer sequences used for RT-PCR and RT-qPCR amplification of human genes.

Gene	Sequence 5′-3′ (F: Forward, r: Reverse)	Product Size (bp)	Annealing Temperature (°C)
*hsRAGE*	F: TGTGCTGATCCTCCCTGAGA	139	61
	R: TGCAGTTGGCCCCTCCTCG		
*hs36B4*	F: AGGCTTTAGGTATCACCACT	219	61
	R: GCAGAGTTTCCTCTGTGATA		
*hsRSP17*	F: TGCGAGGAGATCGCCATTATC	169	61
	R: AAGGCTGAGACCTCAGGAAC		
*hsMyD88*	F: GCAGGAGGAGGCTGAGAAGC	177	66
	R: CGGATCATCTCCTGCACAAACT		
*hsTIRAP*	F: AAGTACCAGATGCTGCAGGCC	200	66
	R: AGTGTCAACTGAGTGTCTGCAG		
*hsDia-1*	F: AGAGCCACACTTCCTTTCCATC	167	66
	R: TCAATCTCAATCTGGAGGTGCC		
*hsIL6*	F: AATGAGGAGACTTGCCTGGTG	143	61
	R: AGGAACTGGATCAGGACTTTTG		
*hsIL8*	F: TGATTTCTGCAGCTCTGTGTG	154	61
	R: TCTGTGTTGGCGCAGTGTGG		
*hsIL1β*	F: AATCTCCGACCACCACTACAG	174	62
	R: TCCCATGTGTCGAAGAAGATAG		
*hsMMP9*	F: ATTGACGACGCCTTTGCCCG	201	61
	R: ATGGGCGTCTCCCTGAATGC		
*hsMMP2*	F: AGCTCATCGCAGATGCCTGG	199	61
	R: AAGGGCCTGTGGGAGCCAG		

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
