# Peer review of "Cigarette Smoke Condensate Exposure Induces Receptor for Advanced Glycation End-Products (RAGE)-Dependent Sterile Inflammation in Amniotic Epithelial Cells"

_ijms, 2021, doi:10.3390/ijms22158345_

Round 1

Reviewer 1 Report

General Comments:

This is a well-conceived paper that tests and important hypothesis. The experiments add to our overall understanding of the effect of cigarette smoking on the end of pregnancy. The fetal membranes are an understudied tissue that has a large impact on pregnancy outcomes and thus this study is meritorious not only in its contribution to our understanding of RAGE but also human fetal membrane weakening. 

Specific Comments:

The paper would be considerably improved by review by a native English speaker. There are a few grammatical and punctuation errors, and missing important words. In addition, there are numerous incidences where the word choice is inappropriate or atypical, making the manuscript difficult to follow in some places.

The Introduction is concisely presented, illustrating both the need for the work and how the authors construct their hypothesis. The results narrative is difficult to follow because the results are, in places, not fully described and the numerical results are missing from the appropriate sections. For example, the data in Figure 1 is not numerically described in the results text, orders of magnitude can only be determined by studying the figure. 

There are large black dots on some of the histograms and it is not clear what these are. For example Figure 5A in the MMP9 CSC (above the box blot and significance stars).

Figure 1 comments - the histogram under the Western example gel is the quantification, please make this clear as it is not stated. It is not stated what the arrows are indicating in figure 1D. I assumed it was cytoplasmic label of the acetylated HMGB1 that has made its way out of the nucleus.

Methods:

Which transfection reagent was used for the NFKB Luciferase experiments?

How was the PCR quantification performed? by Delta, Delta CT analysis?

Discussion:

Contains all the important and relevant discussion points. Similarly to other sections of the paper, language improvement would make it much easier to follow. Also the conclusion that something has been 'proven' by a limited number of experiments, is a very strong statement. This could absolutely be a semantic decision and therefore the authors should re-consider how definitive some statements should be.

Author Response

Author's Reply to the Review Report (Reviewer 1)

~ General Comment:

This is a well-conceived paper that tests and important hypothesis. The experiments add to our overall understanding of the effect of cigarette smoking on the end of pregnancy. The fetal membranes are an understudied tissue that has a large impact on pregnancy outcomes and thus this study is meritorious not only in its contribution to our understanding of RAGE but also human fetal membrane weakening. 

The authors thank the reviewer 1 for her/his interest for the work and the manuscript.

~ Specific Comments:

  • The paper would be considerably improved by review by a native English speaker. There are a few grammatical and punctuation errors, and missing important words. In addition, there are numerous incidences where the word choice is inappropriate or atypical, making the manuscript difficult to follow in some places.

The authors are surprised since they sent the original version of the article, before the submission to IJMS to Scribendi (https://www.scribendi.com), for a commercial proofreading. The authors also note that the reviewer 2 did not ask an English improvement. Nevertheless, they note the comment on the English and sent the scientifically revised version to Scribendi another time for English proofreading. They modify the version according to their comments and remarks. The modifications about English language are underlined in Cyan; those to answer to scientific comments in yellow.

  • The Introduction is concisely presented, illustrating both the need for the work and how the authors construct their hypothesis. The results narrative is difficult to follow because the results are, in places, not fully described and the numerical results are missing from the appropriate sections. For example, the data in Figure 1 is not numerically described in the results text, orders of magnitude can only be determined by studying the figure. 

As requested, the authors added the mean ratio in results text for the Figure 1:

Page 2, l.79: (ratio of 2.9)

Page 2, l81: (ratio of 6.6)

Page 2, l82: (ratio of 0.9)

Page 2, l86: (ratio of 2.0)

The authors also added the mean ratio in results text for the Figure3:

Page 4, l126: (ratio of 1.6)

Page 4, l127: (ratio of 1.3)

Page 4, l129: (ratio of 2.4)

Page 4, l130: (ratio of 1.7)

The authors also added the mean ratio in results text for the Figure 4:

Page 5, l153: IL1β (ratio of 10.13), IL8 (ratio of 904.3)

Page 6, l154 : (ratio of 291.103)

Page 6, l155 : (ratio of 45.1)

Page 6, l157 : (ratio of 142.103), but it has no impact on IL1β fold change (ratio of 13.33)

Page 6, l 160: IL1β (ratio of 2.0 at 48h), IL8 (ratio of 4.3 at 48h and 13.6 at 72h), and IL6 (ratio of 2.7 at 72h)

Page 7, l161: for IL8 (ratio of 6.1)

Page 7, l162: IL6 (ratio of 1.6) and (ratio of 1.8).

The authors also added the mean ratio in results text for the Figure 5:

Page 7, l178: ratio of 8.5

Page 7, l179: ratio of 0.7

Page 7, l180: (ratio of 4.8)

Page 7, l181:  (ratio of 0.6).

Page 7, l183-184: (ratio of 1.14)

Page 7, l184 :  (ratio of 0.94)

The authors notified this global point on page 14 (Material and methods section), l464: Mean ratio (treated/control groups) values are indicated in the results part.

  • There are large black dots on some of the histograms and it is not clear what these are. For example Figure 5A in the MMP9 CSC (above the box blot and significance stars).

The authors described the signification of these black dots on the figures legend for Figure 4 and 5: they indicate values outside of the Tukey boxes.

Page 6, l171 and Page 7, l194:  Black dots indicate values outside of the Tukey boxes.

  • Figure 1 comments - the histogram under the Western example gel is the quantification, please make this clear as it is not stated. It is not stated what the arrows are indicating in figure 1D. I assumed it was cytoplasmic label of the acetylated HMGB1 that has made its way out of the nucleus.

The authors added on the Figure 1 legend, Page 3, l94-95: the results quantification are reported on the histogram.

For the white arrows, the authors indicated in the Figure 1 legend l.100: White arrows indicate the cytosolic cloud of HMGB1. Indeed, acetylation of HMGB1 is associated to its translocation but they preferred to not discuss this point in the results, because the antibody used against HMGB1 recognizes HMGB1 deacetylated and acetylated.

Methods:

  • Which transfection reagent was used for the NFKB Luciferase experiments?

The authors added this technical precision in Material & Methods section, Page 13, l443 to 445: Transfections were performed using 3.75 µl of Lipofectamine 3000 (Fisher Scientific) and 2 µl of Reagent P3000 (Fisher Scientific) per well.

  • How was the PCR quantification performed? by Delta, Delta CT analysis?

The authors used standard curve (obtained thanks to the cloning of the gene amplification product) to quantify the expression of amplified transcripts, they added this technical precision in the Material and Methods section, page 13, l.416: Standard curves were used to quantify the amount of amplified transcripts. Results were normalized to the geometric mean of the human housekeeping genes RPL0 (36b4) and RPS17 (acidic ribosomal phosphoprotein P0 and ribosomal protein S17, respectively), as recommended by the MIQE guidelines [54]. 

Discussion:

Contains all the important and relevant discussion points. Similarly to other sections of the paper, language improvement would make it much easier to follow. Also the conclusion that something has been 'proven' by a limited number of experiments, is a very strong statement. This could absolutely be a semantic decision and therefore the authors should re-consider how definitive some statements should be.

As answered to the general comments, the authors improved English thanks to a new submission of the manuscript to Scribendi, a commercial company specialized in proofreading of scientific manuscripts.

The authors moderated the conclusion: Page 9, l292: our work makes clear a way that how cigarette smoking.

Reviewer 2 Report

Summary:  This study focuses on determining if cigarette smoke concentrate (CSC) induced fetal membrane weakening through RAGE mediated sterile inflammation response. The authors postulate that CSC causes amnion cells to increase HMGB1 expression. The authors then conclude that the increase in HMGB1 causes an increase in RAGE activation and RAGE associated signaling proteins, specially MyD88 and NF-κB, which was aborted with the treatment of RAP, an inhibitor of RAGE. The authors further propose that the increase in cytokine markers noted in CSC treatment was a result of increased RAGE signaling due to RAP treatment reducing the increase in cytokine markers. Furthermore, extracellular matrix remodeling was shown to increase with CSC treatment and be reduced with RAP treatment, and therefore the authors concluded that CSC induces RAGE-mediated extracellular matrix remolding.

Major

  1. It’s stated the CDC caused HMGB1 to translocate from the nucleus to the cytoplasm. However, the data presented in Figure 1D is not convincing. While there may be more HMGB1 in the cytoplasm, the pictures look overexposed which makes me wonder if the HMGB1 in the cytoplasm is background/bleed through and not actually HMGB1. Also, with the intensity of Hoescht (red) and HMGB1 (yellow) I would anticipate the merge image to have more orange coloration and it does not, why is this?

  1. Figure 3A indicates to me that CSC is not working through RAGE to mediate any changes. 1) there is no increase in RAGE expression, which would be anticipated due to RAGE expression being regulated by a positive feedback loop and 2) 2 out of 3 genes linked to RAGE did not change with CSC or RAP treatment, indicating CSC may not mediate changes through RAGE signaling but possibly another signaling cascade, such as Toll-like Receptors (TLRs). Due to the close nature of RAGE and TLR and both being able to impact inflammation, it is necessary to examine the expression of TLRs.

  1. While the statistical tests used for the analysis are stated in the methods and figure captions, it is not stated which groups are being compared to one another in the Dunnett’s post hoc. For example, in Figure 3A you indicate significant difference in MyD88 mRNA expression but which group is it significantly different from?

  1. The conclusion drawn from Figure 3 indicates that CSC treatment caused an increase in NF-κB activity and treatment with RAP aborted this increase. The results from the luciferase assay are not completely convincing due to the large overlap in data between the CSC and CSC-RAP treatment groups. Also, I would argue that the immunofluorescent data does not indicate increased NK-κB p65 activity because it is well established that activate NF-κB translocate to the nucleus to alter transcription. The data in this figure shows concentration of NF-κB near the nucleus but I would argue that this data does not indicate NF-κB activation since NF-κB p65 is not located in the nucleus.

  1. The authors state that the impact of CSC on cells is due to RAGE activation and they draw this draw this conclusion based on data using the inhibitor RAP. However, RAGE is not the only target of RAP. More evidence is needed to establish the link between CSC and RAGE. Using another RAGE inhibitor, such as soluble RAGE, should be used to confirm that the impact of RAP is mediated through inhibiting RAGE and not another receptor.

  1. Figure 4 shows that CSC treatment caused an increase in inflammation, but I would argue that treatment with RAP did not reduce the impact of CSC on inflammation. These results appear to support the idea that CSC may be working through another pathway other than RAGE to induce inflammation. While the RAP treatment groups are not as or are not significantly different from, I assume, the control group there is a lot of overlap with the RAP-CSC group and the CSC only group which makes it difficult to believe the conclusions being drawn from these data.

  1. The authors indicate that CSC treatment caused an increase in HMGB1 expression and postulate that this may be the cause for increase inflammation. However, there is no data showing the impact of HMGB1 on tissue/cells. An experiment showing the impact of HMGB1 is necessary for the conclusion that the changes noted in later figures is a result from HMGB1 activating RAGE.

Minor

  1. The sentence on lines 64-66, I believe, should start with “This” and not “It”. If “It” is the correct word choice then the sentence needs to be edited for clarity.

  1. I suggest editing the figures for better clarification. For example, on Figure one you have the tissue indicated under the X-axis and near the X-axis label. It would be clearer to the reader to have the tissue/cell indicators labeled above the plots

  1. Not all antibodies used in the study are listed in the method section.

  1. Figure 1 label does not match the result represented in the image. CSC does not have an adverse effect on choriodecidua.

  1. Why were non-parametric statistical test used for data analysis? Please provide rationale in the method section.

  1. Included the size of the ladder band(s) in Figure 2A because this is needed for the readers to assess the size of the band correlating to a specific gene.

  1. When stating a treatment caused a significant impact, please indicate if the changes was increased or decreased. For example on line 118 it’s stated that “CDC induced significantly the Myd88…” this statement could be interpreted in different ways.

Author Response

Author's Reply to the Review Report (Reviewer 2)

Summary:  This study focuses on determining if cigarette smoke concentrate (CSC) induced fetal membrane weakening through RAGE mediated sterile inflammation response. The authors postulate that CSC causes amnion cells to increase HMGB1 expression. The authors then conclude that the increase in HMGB1 causes an increase in RAGE activation and RAGE associated signaling proteins, specially MyD88 and NF-κB, which was aborted with the treatment of RAP, an inhibitor of RAGE. The authors further propose that the increase in cytokine markers noted in CSC treatment was a result of increased RAGE signaling due to RAP treatment reducing the increase in cytokine markers. Furthermore, extracellular matrix remodeling was shown to increase with CSC treatment and be reduced with RAP treatment, and therefore the authors concluded that CSC induces RAGE-mediated extracellular matrix remolding.

The authors thank the reviewer 2 for her/his interesting remarks that we answered in this point-by-point text and/or we included in the revised version, to ameliorate the manuscript.

Major

  1. It’s stated the CDC caused HMGB1 to translocate from the nucleus to the cytoplasm. However, the data presented in Figure 1D is not convincing. While there may be more HMGB1 in the cytoplasm, the pictures look overexposed which makes me wonder if the HMGB1 in the cytoplasm is background/bleed through and not actually HMGB1. Also, with the intensity of Hoescht (red) and HMGB1 (yellow) I would anticipate the merge image to have more orange coloration and it does not, why is this?

Understanding the comment of the reviewer 2, the authors proposed a revised Figure 1D with a new lower exposition using modified acquisition’ parameters. As previously described, HMGB1 is a protein with a high rate of expression, explaining why the merge image do not give more orange staining. Indeed to support this statement of an important cellular expression, the authors using quantitative RT-PCR assays found Cp around 18 for HMGB1 in control tissues, similarly to Cp obtained for classical housekeeping genes. Moreover, the cytosolic cloud of HMGB1 found in the current literature is not very intense (as obtained by the authors) in tissue with immunohistochemistry (see for example, Lee et al., J Cell Mol Med 2020, Cigarette smoke‐induced HMGB1 translocation and release contribute to migration and NF‐κB activation through inducing autophagy in lung macrophages for example). In addition, literature already identified HMGB1 release by CS smoke (Cheng et al. Mol Bio Cell, HMGB1 translocation and release mediate cigarette smoke–induced pulmonary inflammation in mice through a TLR4/MyD88-dependent signaling pathway) leading our results coherent concerning HMGB1 release in amnion by CSC with a cellular sequence and trajectory : nucleus, cytoplasm and extracellular release.

  1. Figure 3A indicates to me that CSC is not working through RAGE to mediate any changes. 1) there is no increase in RAGE expression, which would be anticipated due to RAGE expression being regulated by a positive feedback loop and 2) 2 out of 3 genes linked to RAGE did not change with CSC or RAP treatment, indicating CSC may not mediate changes through RAGE signaling but possibly another signaling cascade, such as Toll-like Receptors (TLRs). Due to the close nature of RAGE and TLR and both being able to impact inflammation, it is necessary to examine the expression of TLRs.

It is true, that previous works described sometimes a positive feedback of the RAGE expression after its stimulation by ligands (for example as reported by Robinson et al. AJP Lung Cell Mol Phys, 2012 RAGE signaling by alveolar macrophages influences tobacco smoke-induced inflammation). Nevertheless, our group is working on the implication of RAGE in different tissues’ context: in the fetal membranes cells but also in pulmonary and corneal cells with stimulation of RAGE pathway using different ligands (AGEs, HMGB1 and other alarmins); and the current results did not reveal increase of RAGE expression (as for other groups). In the case of this manuscript, it could be explained by the fact that the positive feedback of RAGE is more a consequence of a chronic stimulation. In our case, the treatment is more an acute activation.

Concerning the adaptors, before this work, the induction of the adaptors by RAGE stimulation was never described, leading to a new discovery presented in this paper. It is clearly admitted that RAGE pathway is tissue-specific, so it is very logical that one adaptor is induced, and not the two other ones. This funding could contribute to explain this tissue-specificity.

As discussed later, the authors are convinced that the implication of RAGE is clearly demonstrated by the use of the RAGE-Antagonist-Peptide abbreviated RAP. However, they agree that RAGE and TLRs pathways especially TLR4 pathway are very closed. It is why they already commented this point in the discussion section of the original version, and maintained these lines in the revised version:

Page 8, l235-237: In the literature, we found three major adaptor proteins interacting with the RAGE cytosolic domain and described in inflammatory signal: TIRAP, MyD88 shared with the toll-like receptors TLR2 and TLR4, and Diaphanous-1 [28, 29].

Page 8-9, l259-270: This directly proved that cigarette smoke (CS) could induce inflammation through a RAGE pathway. However, the absence of a total abortion for IL1β release could suggest the intervention of other actors like the well-known TLR4, which can also interact with alarmins. For example, it was found that nicotine modulates the expression of TLR2/4 into cord blood mononuclear cells [41]. Cheng et al. also demonstrated that HMGB1 translocation can mediate CS-induced pulmonary inflammation through the TLR4/MyD88 pathway [42]. However, Allam et al. demonstrated that RAGE and TLR4 differentially regulate the airway responsiveness to cigarette smoke (CS). Indeed, the authors used RAGE and TLR4 knockout mice and found that only RAGE deletion procured protection against CS-induced neutrophila and airway responsiveness [43]. Nevertheless, further studies are required to explore the potential cooperation between these two PRRs concerning sterile inflammation in FM.

In this context, the authors realized, at the beginning of the project leading to this paper, some experiments of RT-qPCR to quantify TLR4 expression in amniotic cells exposed to CSC, and no increase of TLR4 expression was found as for RAGE (n=5, fold change mean at 1.1). Following such results and the specificity of RAP for RAGE inhibition (as described below), we excluded the TLR4 hypothesis proposed by the reviewer 2.

  1. While the statistical tests used for the analysis are stated in the methods and figure captions, it is not stated which groups are being compared to one another in the Dunnett’s post hoc. For example, in Figure 3A you indicate significant difference in MyD88 mRNA expression but which group is it significantly different from?

As stated in the Materials and Methods section and in the legends of the figures, we used one-way ANOVA Kruskal–Wallis test followed by multiple comparison with Dunn’s correction for the comparison of more than two groups. For example in the case of Myd88 expression, we compared the control group to the CSC group and CSC+RAP group. This was the same process for Diaphanous-1 and TIRAP. We could decide to realize different graph for each gene, but we decided to group all the result to a lighter reading.

  1. The conclusion drawn from Figure 3 indicates that CSC treatment caused an increase in NF-κB activity and treatment with RAP aborted this increase. The results from the luciferase assay are not completely convincing due to the large overlap in data between the CSC and CSC-RAP treatment groups. Also, I would argue that the immunofluorescent data does not indicate increased NK-κB p65 activity because it is well established that activate NF-κB translocate to the nucleus to alter transcription. The data in this figure shows concentration of NF-κB near the nucleus but I would argue that this data does not indicate NF-κB activation since NF-κB p65 is not located in the nucleus.

We agree with the reviewer 2, that NFκB induction is not very elevated. However, statistical analysis by ANOVA test revealed induction for the CSC group and not for the CSC+RAP group. The authors observed that the use of this gene reporter assay in these primary amniotic cells never gave very high induction of luciferase activity (weak reactivity of the primary cells). Indeed, we realized some experiments with LPS on the same cells as positive control part (because LPS is a well-known activator of the NFkB pathway), and we obtained similar level’ induction compared to CSC. In addition, even if the authors agree that the staining is not exclusively nuclear, we established that there is a translocation from cytoplasm to nuclear environment, which is in agreement with the activation of the luciferase gene reporter.

  1. The authors state that the impact of CSC on cells is due to RAGE activation and they draw this draw this conclusion based on data using the inhibitor RAP. However, RAGE is not the only target of RAP. More evidence is needed to establish the link between CSC and RAGE. Using another RAGE inhibitor, such as soluble RAGE, should be used to confirm that the impact of RAP is mediated through inhibiting RAGE and not another receptor.

As reviewed in Rojas et al. (Current Drug Target 2019, Inhibition of RAGE Axis Signaling: A Pharmacological Challenge) or in Hudson et al. (Annu Rev MEd 2018, Targeting RAGE Signaling in Inflammatory Disease), many RAGE inhibitors were described in studies: for example blocking peptides as RAP, antibodies against RAGE or synthetic inhibitors. Soluble RAGE was also used to study the RAGE inhibition however this way present some limitations. The major one is that sRAGE inhibit RAGE activity by interacting with RAGE ligands in the extracellular compartment but these RAGE ligands are not all RAGE specific. In this way, the sRAGE inhibition does not also prove the RAGE implication. Moreover, several studies about RAGE activity use in routine this RAP (RAGE-Antagonist peptide) to inhibit specifically the RAGE pathway (see papers with doi : 10.1111/bph.13539, 10.1016/j.ejps.2017.12.019, 10.1158/1078-0432.CCR-12-0221, 10.1038/s41598-019-45798-5, 10.1021/acsbiomaterials.9b00004). Some of these studies used either RAP or sRAGE, or RAP or FPS-ZM1 and obtained the same results in terms of RAGE pathway’s inhibition. It is why the authors do not think that it is necessary to repeat the experiments with another inhibitory strategy. In addition, the authors did not find in the literature that RAP is able to interact with other PRR members’ family, establishing another argument for its specificity to RAGE.

  1. Figure 4 shows that CSC treatment caused an increase in inflammation, but I would argue that treatment with RAP did not reduce the impact of CSC on inflammation. These results appear to support the idea that CSC may be working through another pathway other than RAGE to induce inflammation. While the RAP treatment groups are not as or are not significantly different from, I assume, the control group there is a lot of overlap with the RAP-CSC group and the CSC only group which makes it difficult to believe the conclusions being drawn from these data.

 Our results on Figure 4 describe an induction of IL1B, IL8 and IL6 transcription and release by CSC. Even if at 48h we did not demonstrate RAP action on IL8 release, RAP co-treatment totally abort the release of IL8 and IL6 at 72h, and significantly decreased IL1B release. Obviously, the impact of RAP treatment is not total, but these results clearly demonstrate the implication of RAGE in CSC induced inflammation. Nevertheless, the authors did not assume that RAGE is the only actor, as discussed at the end of the article. In other literature papers working on similar question, it could be observed that RAP treatment leads to a partial blocking, for example in Zhai et al. (Exp Cell Research 2020, The RAGE enhances lung epithelial wound repair: an in vitro study).

  1. The authors indicate that CSC treatment caused an increase in HMGB1 expression and postulate that this may be the cause for increase inflammation. However, there is no data showing the impact of HMGB1 on tissue/cells. An experiment showing the impact of HMGB1 is necessary for the conclusion that the changes noted in later figures is a result from HMGB1 activating RAGE.

The authors totally agree with this comment of the reviewer 2. But, they did not include such results in the original and revised version of this article because they already published last year in Frontier in Physiology, some results about the activation of the RAGE pathway by HMGB1 in amnion and choriodecidua explants (Choltus et al. 2020, Occurrence of RAGE-mediated inflammatory response in human fetal membranes). In this article, they demonstrated that HMGB1 induced the release of some pro-inflammatory cytokines in amnion and choriodecidua and the co-treatment with a RAGE inhibitor decreased this pro-inflammatory cytokines release.

Minor

  1. The sentence on lines 64-66, I believe, should start with “This” and not “It”. If “It” is the correct word choice then the sentence needs to be edited for clarity.

 The authors modified “it” by “this” in the text, l.66.

  1. I suggest editing the figures for better clarification. For example, on Figure one you have the tissue indicated under the X-axis and near the X-axis label. It would be clearer to the reader to have the tissue/cell indicators labeled above the plots

The authors modified the Figure 1 as suggested.

  1. Not all antibodies used in the study are listed in the method section.

 The authors added this technical precision in Materials and Methods section page 13, l.429-431: HMGB1 (1/400, ab79823, Abcam), Myd88 (1/250, ab133739, Abcam), TIRAP (1/100, ab17218, Abcam), Diaphanous-1 (1/400, ab11173, Abcam), p65 NF-κB (1/400, 8242, Cell signaling)

  1. Figure 1 label does not match the result represented in the image. CSC does not have an adverse effect on choriodecidua.

The authors meant that CSC has no impact on choriodecidua in opposition to amnion; we modified the Figure 1 label to be clearer.

Page 3, l.91: CSC treatment induces a cell danger response into amnion.

  1. Why were non-parametric statistical test used for data analysis? Please provide rationale in the method section.

Non-parametric statistical test were used because of the number of samples, which is comprised between 3 and 8, so too small to use parametric tests. The authors checked the distribution, which was not normal and did not allow parametric tests. The authors added this statistical precision Page 14, l.458-459:

Since the results did not follow normal distribution (consequence of the small number of samples), the comparison of means was performed by non-parametric tests

  1. Included the size of the ladder band(s) in Figure 2A because this is needed for the readers to assess the size of the band correlating to a specific gene.

The authors followed the suggestion of reviewer 2 and modified the Figure 2A with size of the ladder band(s). There was a mistake for Myd88 gel: the authors wrote 167pb instead of 177pb. They modified the table 1 too, thank to this comment of the reviewer 2.

  1. When stating a treatment caused a significant impact, please indicate if the changes was increased or decreased. For example on line 118 it has stated that “CDC induced significantly the Myd88…” this statement could be interpreted in different ways.

The authors modified page 4, l.126, “CSC increased significantly the MyD88”. As answered to one comment of the reviewer 1, the authors added the ratio of the fold change in the result text to be clearer.

Reviewer 3 Report

Choltus H et al. evaluated if Cigarette Smoke Condensate exposure induces receptor for Advanced Glycation End-products (RAGE)-dependent sterile 3 inflammation in amniotic epithelial cells.

This study is innovative and important because there is scarce research regarding the effect of tabagism on preterm prelabor rupture of fetal membranes.

In general, it is very well-written, and the experiments are very well performed with no significant flaws.

Author Response

Author's Reply to the Review Report (Reviewer 3)

Choltus H et al. evaluated if Cigarette Smoke Condensate exposure induces receptor for Advanced Glycation End-products (RAGE)-dependent sterile 3 inflammation in amniotic epithelial cells.

This study is innovative and important because there is scarce research regarding the effect of tabagism on preterm prelabor rupture of fetal membranes.

In general, it is very well-written, and the experiments are very well performed with no significant flaws.

The authors thank the reviewer 3 for her/his interest for the work and the positive comments on their manuscript.

Round 2

Reviewer 2 Report

A main concern I had with this manuscript was the results obtained could possibly be due to a receptor other than RAGE, such as a Toll-like receptor. The authors provided a convincing response to this concern and stated preliminary studies lead to the authors not including TLR4 in their hypothesis. I think this information is important and I recommend including this information (RT-qPCR for TLR4) either as a supplementary figure or in a data repository.

Author Response

Comment :A main concern I had with this manuscript was the results obtained could possibly be due to a receptor other than RAGE, such as a Toll-like receptor. The authors provided a convincing response to this concern and stated preliminary studies lead to the authors not including TLR4 in their hypothesis. I think this information is important and I recommend including this information (RT-qPCR for TLR4) either as a supplementary figure or in a data repository.

Answer :The authors thank the reviewer 2 for the positive feedback of our revised version 1. They agree with the reviewer 2 and  add the results of the TLR4 mRNA’ expression by RT-qPCR assays as supplementary figure and propose a related sentence in the discussion section of the new revised version of the  article (underlined in green).

Page 8, l.271-274: In our amniotic context, we tested at the beginning of the project the impact of CSC on TLR4 expression and found that there is no induction (ratio 1.1 by qRT-PCR, supplementary data S1) excluding the TLR4 hypothesis.
